# DISTILLING SAFE LLM SYSTEMS VIA SOFT PROMPTS

## ABSTRACT

Large Language Models (LLMs) have enabled machine learning to be integrated in complex tasks across various domains. This is a cause for concern since LLMs may respond to carefully crafted prompts with unsafe content, thus necessitating concrete safety mechanisms. Current solutions involve dual-model systems combining LLMs with guard models. However, the substantial memory and computational demands of guard models pose significant challenges for deployment. This paper proposes an efficient method for approximating the behavior of dual-model systems using learned embeddings, also known as soft prompts. We introduce a novel distillation framework which optimizes the total variation distance between the outputs of an LLM paired with a guard and the same LLM equipped with our soft prompts. At test time, the learned soft prompts are prepended to user prompts, providing safety at a fraction of the memory and compute costs incurred by a guard model. Our evaluations on various benchmarks demonstrate improved safety of the LLM, offering an efficient alternative to guard models for memory- and computation-constrained settings such as hardware applications.

## 1 INTRODUCTION

Despite their remarkable adoption across research and industry, large language models (LLMs) can generate unsafe and toxic content in response to certain prompts. For example, an LLM might produce harmful or offensive language if manipulated by a malicious user (Xu et al., 2023; Brundage et al., 2018; Liu et al., 2023)

Safety fine-tuning methods such as reinforcement learning (RL), supervised fine-tuning, etc. (Bai et al., 2022), can offer improvements in terms of safety alignment of the base LLM but system-level enhancements and layered defenses are necessary for minimizing risks (Meta, 2024). To address this, guard models (Inan et al., 2023) have been introduced to evaluate and maintain the safety of LLM responses to user prompts. In this design the guard model assesses the safety of the response before exposing it to the user. In a nutshell, a guard model is a separate LLM that classifies the input pair $(x, y)$ as safe or unsafe with $x$ being the user's prompt and $y$ being the response of the LLM. When $(x, y)$ is deemed unsafe a pre-defined refusal answer, such as "Sorry, I cannot help with this matter.", overrides the initial response $y$. This approach is the last line of defense against toxic and harmful responses, while preserving the capabilities of the LLM when its response is deemed safe. While recent studies (Mangaokar et al., 2024) have shown that guard models are vulnerable to adversarial perturbations, they are used as a de-facto method for building safe LLM systems.

The dual-model approach demands significant memory and computational resources, making it especially unsuitable for on-device deployment where memory and compute is limited (Qin et al., 2024). This is illustrated in Figure 1. In addition, the sequential nature of this approach (i.e. the guard waits for the full output of the LLM before classifying it) will degrade important metrics such as time-to-first-token. Various strategies have been proposed to address this issue, including quantizing the models to reduce memory consumption, distilling large LLMs into smaller models, and fine-tuning LLMs to mitigate toxic outputs (Lin et al., 2024; Fedorov et al., 2024). While these methods improve memory efficiency and enhance safety, they often compromise the LLM's generalization capabilities (Xu et al., 2024a).

In this work, we propose a compute and memory efficient LLM system that operates both usefully and safely. In particular, we equip LLMs with an extra set of learned embeddings, known as soft prompts, to approximate the functionality of the dual-LLM system. To do so, we employ a novel

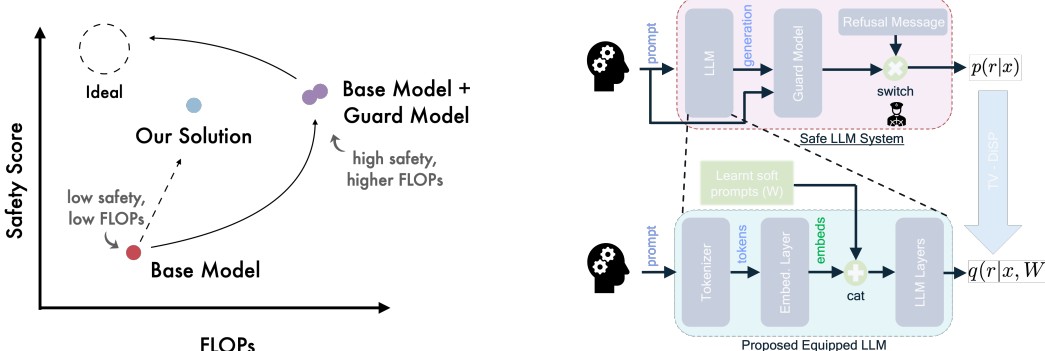

Figure 1: **Left: Safety - Compute Trade-off.** LLMs (denoted as Base Model) can generate unsafe and toxic content. When paired with a Guard Model; altogether called a safe LLM system, their safety improves at the expense of a substantial compute and memory penalty which may hinder their usability. In this work, we distill safe LLM systems into a single model and shrink this penalty. **Right: Pipeline for our proposed TV-DiSP.** We distill a safe LLM system composed of a paired LLM and guard model into a set of learnable parameters equipped to the LLM.

distillation framework through optimizing the total variation distance between the output of the safe LLM system and our enhanced LLM. At test-time, we prepend the learned soft prompts to the user's prompt before feeding them to the LLM. This approach seeks to maintain the safety and efficacy while reducing the computational overhead, thereby making it more feasible to deploy on a wide range of resource-constrained devices. We assess the efficacy of our approach by testing the safety and usefulness of the LLM responses on several benchmarks, architectures, and against other parameter efficient fine-tuning methods, providing consistently better results. Our contributions are thus three-fold:

1. We propose an efficient alternative to dual-model safe LLM systems based on learned soft prompts for the base LLM. In doing so, we obtain safer LLMs that are also more applicable for resource-constrained applications.

2. We propose a novel total variation optimization framework for distilling the behavior of the safe LLM system into a set of soft prompts.

3. We provide a comprehensive evaluation scheme spanning four different LLMs, three different datasets , and three different parameter efficient fine-tuning methods for comparing the various approaches showing the superiority of our proposed method.

## 2 RELATED WORK

**LLM Safety.** Recent studies have highlighted the susceptibility of large language models (LLM) to generating toxic or unsafe content with carefully designed prompts (Mazeika et al., 2024; Chao et al., 2024; Hartvigsen et al., 2022), or when exposed to adversarial attacks (Liu et al., 2023; Gong et al., 2025; Zou et al., 2023). This has motivated researchers to explore various strategies to improve the safety alignment of LLMs. Among the most prominent approaches are Reinforcement Learning with Human Feedback (RLHF) (Dong et al., 2024) and the use of auxiliary guard models (Inan et al., 2023; Padhi et al., 2024). Although these methods have shown promise in enhancing the safety of LLM outputs, they often come with significant drawbacks: RLHF requires costly training pipelines, and guard models can introduce substantial computational overhead during inference. In this work, we propose a parameter-efficient fine-tuning approach that distills the safety benefits of guard models into the base LLM, aiming to retain safety improvements while reducing inference costs.

**Adapting LLMs** Despite the impressive capabilities of recent large language models (LLMs) across a wide range of tasks, they often underperform when dealing with domain-specific knowledge or when their weights are quantized for on-device deployment. To address this performance gap, several parameter-efficient adaptation techniques have been proposed in the literature, including

the widely adopted Low-Rank Adapters (LoRA) (Hu et al., 2022; Dettmers et al., 2023), steering vectors (Turner et al., 2023; Panickssery et al.; Wang & Shu, 2023), circuit breakers (Zou et al., 2024), and the more recent soft prompt tuning approach (Xu et al., 2024a; Zheng et al., 2024). Among these, soft prompt tuning has shown significant promise in preserving model performance both before and after quantization. In this work, we investigate parameter-efficient fine-tuning methods—focusing particularly on soft prompt tuning—as a means to distill the safety capabilities of an LLM system equipped with a guard model back into the base LLM. This enables a more effective and computationally efficient alternative to deploying guard models at inference time.

## 3 METHODOLOGY

**Preliminaries.** Let $p(y|x)$ represent an LLM that generates $y$ in response to a prompt $x$. Further, let $p(s|x, y)$ represent a guard model that generates a safety label $s \in \{0, 1\}$ given the prompt-response pair $(x, y)$ where $s = 1$ represents the label "safe" for the LLM's generation $y$. A safe LLM system consists of both the LLM and guard model $p(y, s|x) = p(y|x)p(s|x, y)$. This system returns to the user a response $r$ whose contents depend on the safety score of the pair $p(s|x, y)$. We formulate the responses from the safe LLM system as

$$p(r|x, y) = p(s = 1|x, y)\mathbb{I}(r = y) + p(s = 0|x, y)\mathbb{I}(r = y_r) \qquad (1)$$

where the $y_r$ is a pre-defined refusal response such as "Sorry, I cannot help with this matter." and $\mathbb{I}(.)$ is the indicator function. The safe LLM system output distribution can thus be formalized as

$$p(r|x) = \sum_y p(y|x)p(r|x, y). \qquad (2)$$

One major downside in deploying such a system is that it requires two full forward-passes through the LLMs (*i.e.* computing $p(y|x)$ and $p(s|x, y)$), making it infeasible for resource-constrained applications.

### 3.1 TV-DISP: DISTILLATION VIA SOFT PROMPTS

In this section, we propose our novel adaptation strategy to distill the safe LLM system (described in Sec. 3) to an instance of the LLM which is equipped with extra learnable parameters. Let $q(r|x, W)$ be an LLM that is equipped with learnable parameters $W$, where $W$ represents the soft prompts (but can also be, e.g., LoRA parameters).

**Total Variation Optimization** The total variation distance is a suitable choice as the primary objective for our distillation because it provides probabilistic guarantees on how far the distilled model can deviate from the distillation target in terms of downstream task performance. More specifically, we present the following theorem where the proof is left for the appendix.

**Theorem 3.1.** *Let $p(r|x)$ be the safe system and $q(r|x, W)$ be the LLM equipped with soft prompts. We have that the performance gap between them on any test function $\phi(\cdot)$ with $|\phi(\cdot)|_\infty \le 1$ is*

$$\left| \mathbb{E}_{p(r|x)}[\phi(r)] - \mathbb{E}_{q(r|x, W)}[\phi(r)] \right| \le 2D_{TV}\left(p(r|x), q(r|x, W)\right),$$

*where $D_{TV}(\cdot, \cdot)$ is the total variation distance.*

Having guarantees is especially desirable for safety-sensitive applications. Theorem 3.1 can apply by considering $\phi(\cdot)$ as the safety probability / binary decision given by a model and/or human.

As previously mentioned, we focus on the case where the learnable parameter $W$, i.e. the outcome of the distillation process, represent soft prompts. Once we have distilled the safe LLM system into these soft prompts $W$, they are prepended to the sequence of token embeddings of the user prompt and are fed into subsequent layers. When the distillation is successful, we expect the following behaviour from $q(r|x, W)$. For safe responses, (*i.e.* $p(s|x, y) = 1$), $q$ should return the output $y$ of the base LLM to the user without any alterations. This helps to preserve the utility of the underlying LLM. Otherwise for unsafe responses, (*i.e.* $p(s|x, y) = 0$), $q$ should return the pre-defined refusal message. By satisfying these two cases, our distilled $q$ recovers the full functionality of the safe LLM

system $p(r|x)$. We optimize the learnable parameters $W$ to minimize the total variation distance between the two distributions $p(r|x)$ and $q(r|x, W)$ as follows:

$$W^* = \arg \min_W \mathbb{E}_x \left[ D_{TV} \left( p\left(r|x\right), q\left(r|x, W\right) \right) \right],$$

where we have that the TV distance can be upper bounded as

$$D_{TV}(q, p) \le 1 - \mathbb{E}_{p(y|x)p(r|x,y)} \left[ \min \left( \frac{q(r|x, W)}{p(r|y, x)}, 1 \right) \right] \tag{3}$$

While optimizing $D_{TV}$ is aligned with our objectives, the loss defined in Equation 3 can be hard to optimize due to operating on probabilities directly. It is thus easier to optimize the following which relies on log-probabilities instead

$$\max_W \mathbb{E}_{p(y|x)} \left[ p(s=1|x,y) \left[ \log \frac{q(r=y|x, W)}{p(s=1|x,y)} \right]_- + p(s=0|x,y) \left[ \log \frac{q(r=y_r|x, W)}{p(s=0|x,y)} \right]_- \right], \tag{4}$$

where $p(s=1|x,y)$ and $p(s=0|x,y) = 1 - p(s=1|x,y)$ are the probabilities that $(x, y)$ is safe and unsafe respectively, and $[z]_- = \min(z, 0)$. The first term in Equation 4 preserves the LLM response when it's deemed safe by the guard model while the second term learns the refusal message for unsafe responses. Training $W^*$ in this fashion only requires a dataset of prompts without labels as the guard model dictates whether each prompt is safe or unsafe. We denote our method Total Variation-based Distillation via Soft Prompts as TV-DiSP.

**Inference.** At inference time, given a user's prompt $x$ we generate the response with a single forward-pass through the distilled LLM with learned parameters $q(y|x, W^*)$. Note that in this forward-pass, the added compute and memory requirements for a moderately-sized $W^*$, e.g. 100 soft prompt vectors, are substantially lower than to what is required for two forward-passes when computing $p(y|x)$ and $p(s|x, y)$ in Equation 1. Through our experiments we will show that even a small $W^*$, e.g. 100 soft prompts consisting of a few thousand parameters, is sufficient to reduce the total variation distance to a sufficiently small value which maintains the fluency of the LLM and the safety provided by the guard model.

## 3.2 Other Optimization Schemes for Safety Alignment

Besides our proposed TV-distillation scheme, we explore the efficacy of other loss functions for this purpose. In particular, we explore three strong baselines as competitors:

**Perplexity Optimization.** Recently, Xu et al. (2024a) demonstrated how soft prompts can be trained to alleviate quantization-induced performance degradation by directly optimizing perplexity. As a baseline we explore perplexity optimization as an alternative to the total variation distance. This entails learning $W$ by optimizing the perplexity of $q$ on a given dataset. Formally, and following our notation, the perplexity optimization solves the following optimization problem:

$$\min_W \ -\log q(r = x_{t+1}|x_{1:t}, W).$$

Note that this baseline follows the next token prediction (*i.e.* $x_{t+1}$) when observing the $t$ tokens from the sequence ($x_{1:t}$).

**REINFORCE.** Next, we analyze an alternative baseline that optimizes for the safety score directly. We follow the standard practice in the reinforcement learning literature by applying the log trick (REINFORCE) to calculate a tractable gradient through the following formulation:

$$\max_W \ \mathbb{E}_{q(y|x,W)} \ p(s = 1|x, y)$$

This baseline directly optimizes for the safety score of the model, measured by the guard model.

**KL-Distillation.** At last, we replace our proposed TV approach with minimizing the Kullback-Leibler distance between the safe LLM system, and the LLM equipped with $W$ through

$$\min_{W} \quad \mathbb{E}_x \left[ D_{KL} \left( p\left( r|x \right), q\left( r|x, W \right) \right) \right].$$

We can show that this specific loss also provides guarantees on the downstream behavior, albeit looser than the ones we obtain with the total variation loss. More specifically, through an application of Pinsker's inequality (Csiszár & Körner, 2011), we have the following simple upper bound

$$\left| \mathbb{E}_{q(r|x,W)}[\phi(r)] - \mathbb{E}_{p(r|x)}[\phi(r)] \right| \leq 2D_{TV}\left( q(r|x,W), p(r|x) \right) \leq \sqrt{2D_{KL}(q(r|x,W), p(r|x))}.$$

Therefore, the total variation loss is better in capturing differences in downstream performance compared to the KL divergence, something which we will observe empirically in our experiments.

### 3.3 EXTENSION TO OTHER PEFT APPROACHES

In previous sections, we assumed that $W$ is a set of soft prompts prepended in the embedding space to the user's prompt. Nonetheless, our formulation is generic to be applied other Parameter Efficient Fine-Tuning (PEFT) methods such as Low Rank Adaptors and Steering Vectors.

**Low Rank Adaptors (LoRA).** LoRA (Hu et al., 2022) introduces trainable low-rank matrices that are injected into the attention and/or feed-forward layers of the transformer architecture. In our framework, the optimization objective over $W$ can be reinterpreted as learning these low-rank adapters, where the safety-aligned behavior is induced by constraining the latent representations via our proposed regularization. This allows LoRA to inherit the safety properties of soft prompt tuning while maintaining its parameter efficiency. To ensure a fair comparison with soft prompt tuning, we set the rank of the LoRA adapters such that the total number of learnable parameters matches that of the soft prompts.

**Steering Vectors (SV).** Steering vectors (Turner et al., 2023) operate by linearly modifying the hidden states of the model to induce specific behaviors. Our method can be adapted to learn such vectors by treating W as a directional offset in the embedding or hidden space. The safety alignment is achieved by optimizing W to steer the model's responses toward desired safety criteria, effectively embedding behavioral constraints directly into the latent dynamics.

## 4 EXPERIMENTS

### 4.1 SETUP AND EVALUATION PROTOCOL

**Models.** In our experimental setting, we focus on mimicking the on-device setting for when LLMs are deployed on edge devices. To that regard, we run all our experiments by quantizing the weights of all models to 4-bits using the optimum-quanto library. We experiment with four different models including Qwen2-1.5B (Bai et al., 2023), Gemma2-2B (Team et al., 2024), Llama3-instruct-1B, and Llama3-instruct-3B parameters (Fedorov et al., 2024). Given the resource constraints typical of edge AI platforms, we selected smaller language models that are instruction-tuned and strike a good balance between performance and computational efficiency. Furthermore, we use LlamaGuard3-1B as the guard model that provides the safety (*i.e.* $p(s|x,y)$) score for distillation training and LlamaGuard3-8B to evaluate the distilled models (Inan et al., 2023).

**Evaluation Metrics.** Since this work aims at studying safety-based LLM systems, we first assess the safety of the generation from the LLM before and after equipping it with the learnt $W$. We leverage the state-of-the-art Llama3Guard-8B (Inan et al., 2023) parameter model to be the evaluator where we report the Safety Guard Score (SGS) defined as:

$$SGS = \mathbb{E}_{x \sim \mathcal{D}}[\mathbb{I}(p(s = 1|x, r) > 0.5)], \tag{5}$$

where $\mathcal{D}$ is the validation set of a given dataset, $x$ and $r$ are the prompt and its corresponding generation from LLM, respectively, and $\mathbb{I}$ is the indicator function. Further, we compare the memory and computational needs to run different approaches such as the base LLM, the safe LLM system, and the LLM equipped with $W$. In terms of computation, we report the FLOPs needed to generate a single

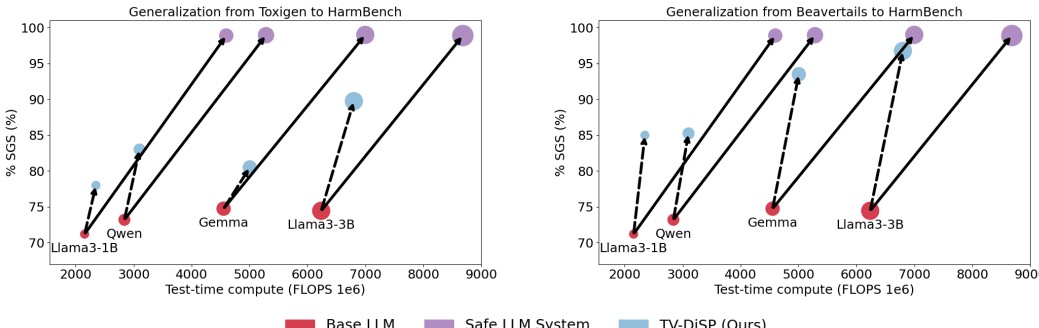

Figure 2: **Safety-Compute trade-offs when trained on Beavertails or Toxigen, and tested on HarmBench.** We report on the y-axis the Safety Guard Score (SGS) according to LlamaGuard3-8B for three variations: the base LLM (red), the safe LLM system with LlamaGuard3-1B in-the-loop (purple), and our proposed distilled LLM with soft prompts (blue). The x-axis shows the test-time compute measured in the number of floating-point operations (FLOPs) to generate a single token for a context length of 512 on a fixed batch of data. The size of the circles represent the relative memory requirement for each variation. Our proposed TV-DiSP succeeds in distilling the safety of the safe LLM system with significant less memory and computation requirement.

token under a fixed context length of 512 (we leave to the appendix results under larger context length). Further, we complement our evaluation paradigm to include measuring the usefulness of the LLM upon equipping it with the learned $W$. To do so, we conduct the standard 5-shot MMLU (Hendrycks et al., 2020) evaluation and report the accuracy of the model as a usefulness metric.

**Datasets.** Regarding the datasets, we experiment with training on the Beavertails (Ji et al., 2023) dataset, where we subsample a fixed set of $10k$ prompts. Further, and to assess the generalizability of our approach, we also leverage the standard toxigen (Hartvigsen et al., 2022) dataset that includes both toxic and non-toxic prompts for training $W$. In particular, we randomly subsample a fixed set of $5k$ prompts from the dataset and use them for the training experiments. It is worth mentioning that in all our experiments, we conduct a single epoch of training (the model trains on each data point only once) for efficiency purposes. To assess the reliability of the learned $W^*$, we conduct our safety evaluation on an out-of-distribution setting. In particular, we experiment with the standard benchmark HarmBench (Mazeika et al., 2024), a collection of harmful adversarial prompts. Moreover, we also include evaluations on Detect-JailBreak; a collection of three different datasets used for LLM safety evaluation (Shen et al., 2024; Xu et al., 2024b; Li et al., 2024; Zou et al., 2023). At last, we leverage the test-set of Beavertails to include in-domain performance evaluation. By leveraging these datasets, we provide a comprehensive evaluation scheme. Remaining of training details are in the appendix.

### 4.2 TV-DiSP: RECOVERING SAFETY WITH DISTILLATION

We first assess the efficacy of our proposed TV-DiSP in distilling the performance of a safe LLM system composed of the base LLM and the guard model. Figure 2 reports the results where the $x$-axis reports the computational requirements in FLOPs, the $y$-axis reports the safety guard score (SGS), and the diameter of each reported circle represents the relative memory requirement to store the deployed model on-device. For this experiment, we analyzed 4 different LLMs namely; Llama3-1B, Qwen2-1.5B, Gemma2-2B, and Llama3-3B instruct tuned models. We train the soft prompt on either Toxigen (left figure) or Beavertails datasets (right figure), where red, purple and blue circles represent the base LLM, the safe LLM systesm (LLM + Llama Guard 1B), and our proposed TV-DiSP. In this experiment, we set $W$ to be a set of 100 soft prompts.

We observe **(i)** The safe LLM system can indeed identify unsafe generations by the LLM and correct them to a refusal response. For example, the SGS of Llama3-insruct-1B model improves from 71% to 99%, measured by LlamaGuard-8B. However, this safety gain comes at a big expense in both memory and computation. For example, generating a single token from the base model requires $2.1 \times 10^9$ flops whereas the safe system requires $4.6 \times 10^9$ flops and doubles the memory requirements. **(ii)** TV-DisP can successfully distill the safe LLM system into a single model equipeed with additional

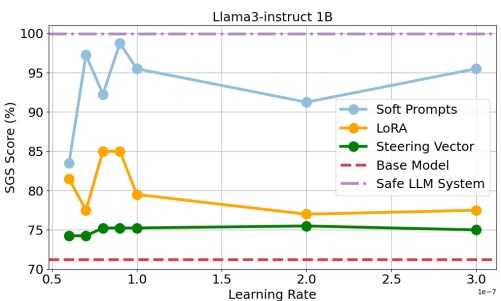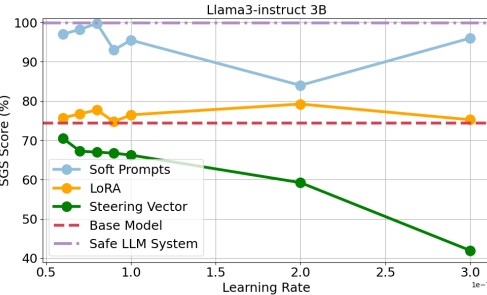

Figure 3: **Comparing TV-DiSP against TV-DiSV and TV-DiLoRA.** We employ our distillation scheme in Equation equation 4 to distill the safe LLM system into a steering vector (SV) or a low rank adaptor (LoRA). We conduct a single epoch training on Beavertails under different learning rates and report SGS on HarmBench. TV-DiSP consistently outperforms TV-DiSV and TV-DiLoRA.

learnt embeddings. For example, when $W^*$ is trained on Beavertails, TV-DiSP improves the safety of the base LLM by 20% with less than 10% additional computational cost, and less than 1% additional memory consumption on both Gemma and Llama3-3B models. It is noteworthy to mention that our experiments follow a challenging evaluation protocol by evaluating on out-of-distribution (mismatch between training and testing datasets). That is, during the distillation phase, the model did not observe any adversarial prompts, similar to the ones in HarmBench. This further strengthens the reliability and generalizability of the provided results. **(iii)** Different training distribution can result in variation of the attained performance gain by TV-DiSP. The is exemplified by changing the training distribution from Beavertails to Toxigen and conducting the same distillation scheme. While TV-DiSP still provides consistent safety gains when compared to the base LLM, this performance improvement is enlarged with the better training distribution of Beavertails. To that regard, in the rest of our experimentation in the paper, we conduct training with the stronger Beavertails dataset.

### 4.3 COMPARISONS TO LORA AND STEERING VECTORS

Next, we set to study the efficacy of soft prompts as a parameter efficient fine-tuning method for distilling safe LLM system as compared to LoRA and Steering Vectors, dubbed as TV-DiLoRA and TV-DiSV, respectively. To do so, we employ our total variation distillation scheme described in Section 3.1. For LoRA adapters, we set the rank to match the number of learnable parameters in the case of 100 soft prompts. To alleviate the impact of training with sub-optimal learning rate for each method, we conduct the training on Beavertails with 7 different learning rates $[6, 7, 8, 9, 10, 20, 30] \times 10^{-4}$ and report the SGS on HarmBench in Figure 3 for LLama3-1B and Llama3-3B models. The dashed lines represent the performance of the base model and the safe LLM system.

We report **(iv)** Across al learning rates, soft prompts provide consistently the largest safety gains compared to LoRA adapters and Steering Vectors under both considered models. In fact, the performance between TV-DiSP and TV-DiLoRA can grow larger than 20%, as measured by the Llama-Guard-8B model. **(v)** Our total variation distillation is a generally effective distillation scheme, and not applicable to just soft-prompt learning. This is demonstrated with the safety gains that TV-DiLoRA provides when compared to the base LLM. **(vi)** Steering vectors do not have enough capacity to distill the guard model providing mixed performance; marginal improvement is observed in the Llama3-1B case, but performance deterioration is recorded in the Llama3-3B case.

### 4.4 TV-DISP: BETTER TRADEOFF FOR SAFETY VS USEFULNESS

Given the strong potential of distilling safe LLM systems into a few learnable embeddings,*i.e.* soft prompts, we study the impact of different distillation objective functions. In particular, we explore learning $W$ with three other objective functions: perplexity optimization (Perplexity), optimizing the safety score directly through policy gradient (REINFORCE), and KL divergence optimization. Please refer to Section 3.2 for mathematical formulation details. Similar to our setup in Section 4.3, we analyze two LLMs: Llama3-instruct 1B and 3B models, and train on Beavertails dataset.

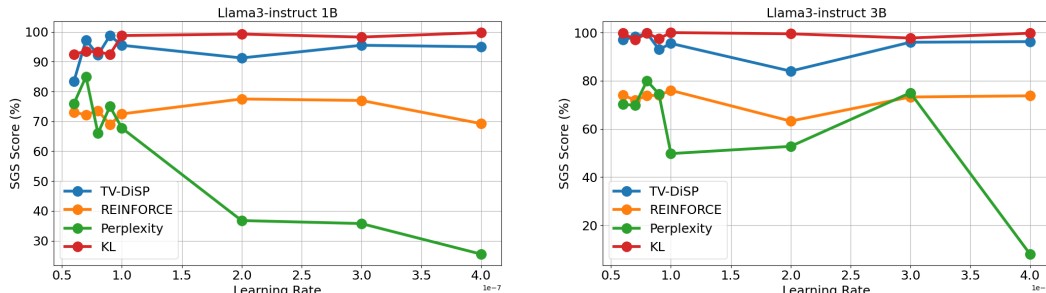

Figure 4: **Comparing TV-DiSP against other distillation schemes.** We compare our proposed total variation objective function to other loss functions in distilling the safe LLM system. We experiment with perplexity optimization, REINFORCE and KL divergence minimization. We report on the x-axis the learning rate used for training, the SGS on the y-axis on HarmBench. Left: Llama3-1B and Right: Llama3-3B model is the base LLM.

Figure 4 shows the safety curves for each distillation method under different learning rates used in training, to alleviate suboptimal training hyperparameters. We observe **(vii)** perplexity optimization provides small safety improvement under small learning rates. However, under relatively large learning rates, perplexity optimization degrades the SGS due to overfitting to the training distribution. Similar to Perplexity, REINFORCE suffers from poor out of distribution generalization, providing marginal safety improvement on HarmBench. **(viii)** KL distillation succeeds in distilling the safety behavior of the safe LLM system providing comparable SGS scores to the proposed TV approach.

While KL and TV have very comparable SGS, an important question arises: What is the cost on the performance under non-toxic prompts? To address this question, we perform a 5-shot in-context learning evaluation on the standard MMLU (Hendrycks et al., 2020) benchmark and report accuracy as a proxy for usefulness. For each distillation framework, we select the soft prompts that yield the best performance (based on the optimal learning rate) to ensure a fair comparison. Figure 5 illustrates the trade-off between usefulness and safety for the Base LLM, the Safe-LLM system, our proposed TV-DiSP method, and the REINFORCE and KL distillation schemes. We observe that **(ix)** our proposed TV-DiSP achieves the best balance between safety and usefulness, demonstrating a more effective distillation approach than KL. Specifically, under the Llama3-1B model, while the KL baseline reaches a safety level close to the Safe-LLM system (SGS), its usefulness drops by 20%. Finally, these results highlight the need for even stronger distillation strategies that can further improve safety without compromising usefulness relative to the Base LLM.

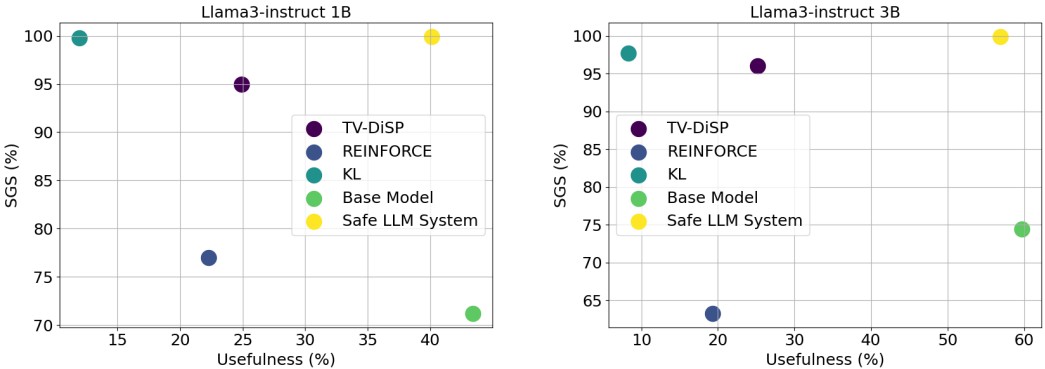

Figure 5: **Comparing TV-DiSP against other distillation schemes in terms usefulness vs safety.** The x-axis reports the usefulness: 5-shot in context learning accuracy on MMLU benchmark, and the y-axis shows the SGS measured by Llama3 Guard - 8B for our proposed TV-DiSP, against REINFORCE and KL. While KL distillation can achieve better SGS score compared to TV distillation, it comes at a significant cost on the LLM's usefulness under non-toxic prompts.

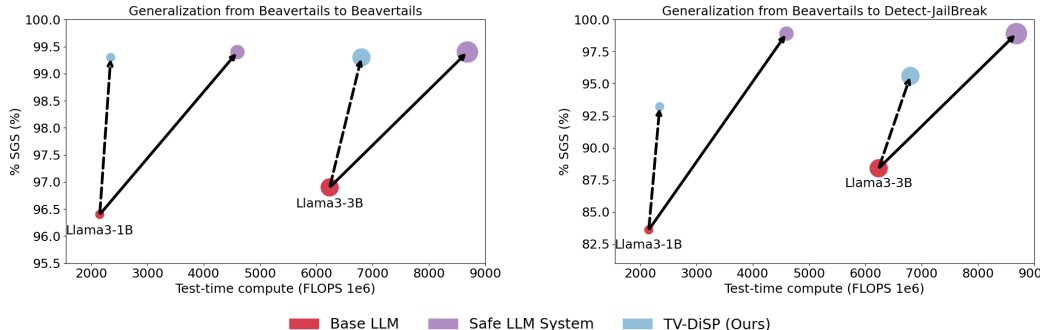

Figure 6: **Generalization to in-distribution and out-of-distribution.** Left: Results on Beavertails test-set (in distribution). Right:Results on Detect-Jailbreak dataset. Our proposed TV-DiSP provides consistent safety gains on both in- and out of- distribution settings on two different LLM architectures.

### 4.5 EXPERIMENTS ON OTHER SAFETY BENCHMARKS

In all our previous experimentation, we focused our evaluation on the standard HarmBench dataset. In this section, we explore the efficacy of our proposed TV-DiSP under two different settings: the easy setting of evaluating in-distribution and the challenging jailbreak setting. For the first setting, we conduct our evaluation on the test-set of Beavertails Ji et al. (2023) dataset. For the second setting, we leverage a subset of the Detect-Jailbreak (Shen et al., 2024; Xu et al., 2024b; Li et al., 2024; Zou et al., 2023) benchmark, which is a collection of three different datasets used to evaluate LLM safety. In particular, we leverage a subset of Detect-JailBreak where all prompts are labeled as jailbreaks. We feed these prompts to Llama3 1B and 3B models and record the SGS measured with Llama3-Guard-8B model. Figure 6 summarizes the results.

We observe: **(x)** Our proposed TV-DiSP provides consistent performance improvement under both scenarios by successfully distilling the safe LLM system. In particular, and under the challenging Detect-JailBreak benchmark, We improve the safety score SGS by more than 5% under two different LLMs. Furthermore, the safety improvement provided by TV-DiSP is also observed on the easier in-distribution setting with a consistent safety enhancement of more than 1%. These results complement our findings in earlier sections on the efficacy of our proposed method and further shows the generalization of our TV-DiSP under different testing settings.

**Section Summary.**   In this section, we conducted a comprehensive experimental evaluation of our proposed TV-DiSP approach in distilling safe LLM systems. We showed the generalizability of our approach under different architectures and training distributions **(i-iii)**, its superiority when compared to other parameter efficient fine-tuning methods **(iv - vi)**, its advantages when compared to other safety optimization schemes **(vii - ix)**, and finally its consistency under different evaluation benchmarks **(x)**. We leave to the appendix further experiments including ablating the impact of changing the number of learned soft prompts (i.e. the size of $W$) and optimizing $W$ with PPO (Schulman et al., 2017).

## 5 CONCLUSIONS

In this work, we introduced a lightweight and efficient alternative to traditional safe LLM system by distilling the behavior of a dual-model architecture (LLM + Guard) into a single quantized LLM augmented with learned soft prompts. Our method minimizes the total variation distance between the outputs of the original safe system and the enhanced LLM, enabling safety-aligned generation without the computational burden of running a separate guard model. This approach significantly reduces the memory and latency overhead, making it suitable for deployment on resource-constrained devices. We validated our framework through experiments on four different LLM architectures including Llama, Qwen, and Gemma models, two different training distributions, two different testing datasets, and against three different safety alignment optimization schemes. Our results demonstrate that the proposed approach consistently and significantly outperforms all evaluated baselines.

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

# A  METHODOLOGY

## A.1  PROOF OF THEOREM 3.1

In section 3.1, we provided a theoretical statement on the probabilistic guarantees that the TV distillation approach provides. In this section, we provide its proof.

**Theorem A.1** (restatement). *Let $p(r|x)$ be the safe system and $q(r|x,W)$ be the LLM equipped with soft prompts. We have that the performance gap between them on any test function $\phi(\cdot)$ with $|\phi(\cdot)|_\infty \leq 1$ is*

$$\left| \mathbb{E}_{q(r|x,W)}[\phi(r)] - \mathbb{E}_{p(r|x)}[\phi(r)] \right| \leq 2D_{TV}\left(q(r|x,W), p(r|x)\right),$$

*where $D_{TV}(\cdot,\cdot)$ is the total variation distance.*

*Proof.* The statement is a direct consequence of the sup representation of the total variation distance (Polyanskiy & Wu, 2014)

$$D_{TV}(q,p) = \frac{1}{2} \sup_{\{\phi,|\phi|_\infty \leq 1\}} |\mathbb{E}_q[\phi] - \mathbb{E}_p[\phi]| \geq \frac{1}{2}|\mathbb{E}_q[\phi] - \mathbb{E}_p[\phi]|,$$

for $\{\phi, |\phi|_\infty \leq 1\}$. □

## A.2  DERIVATION OF EQUATION EQUATION 3

Next, we derive the upper-bound of the total variation distance showed in Equation equation 3. This upper-bound is useful for facilitating the optimization of the total variation distance.

$$D_{TV}(q,p) = \frac{1}{2}\sum_r |q(r|x,W) - p(r|x)|$$

$$= \frac{1}{2}\sum_r \left| \mathbb{E}_{p(y|x)}\left[q(r|x,W) - p(r|x,y)\right] \right|$$

$$\leq \mathbb{E}_{p(y|x)}\left[D_{TV}\left(q(r|x,W), p(r|x,y)\right)\right]$$

$$= 1 - \mathbb{E}_{p(y|x)}\left[\sum_r \min(q(r|x,W), p(r|x,y))\right]$$

$$= 1 - \mathbb{E}_{p(y|x)p(r|x,y)}\left[\min\left(\frac{q(r|x,W)}{p(r|y,x)}, 1\right)\right].$$

# B  EXPERIMENTS

## B.1  SETUP AND EVALUATION PROTOCOL - EXTENDED

In section 4.1, we provided the crucial details for our experimental setup. Given the space limiataiton, and for transparency and full reproducibility, we provide the rest of the details for our setup and evaluation protocol in this section.

**Training Details.**  In all our trainings, we fixed Adam (Kingma & Ba, 2017) to be the optimizer in action with $\epsilon = 10^{-7}$. Regarding LoRA: We set the rank to 2 or 3 that matches the number of learnable parameters in the set of soft prompts. Regarding SV: We apply the learnt steering vector to the output of layer 13, following the standard practices (Panickssery et al.).

**Evaluation Details.**  Regarding the evaluation dataset: for HarmBench, we leveraged all the 400 available prompts in the evaluation. For Detect-Jailbreak, we leverage a subsample of 500 prompts that are both labeled as jailbreak prompts and regularly constructed (not adversarial prompt injection). For the evaluation on Beavertails, we sub-sampled a fixed set of $1k$ prompts from the test set. For the choice of learning rate in Section 4.4, we defined the optimal learning rate to be the one with the best sum of safety and utility (i.e. SGS+MMLU Accuracy).

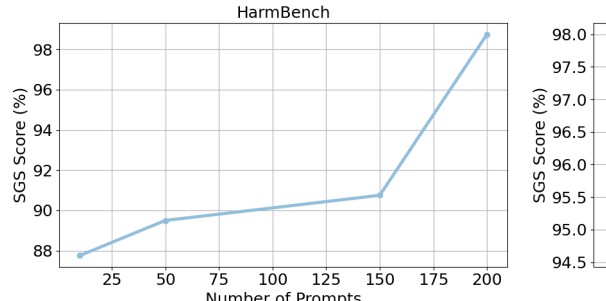 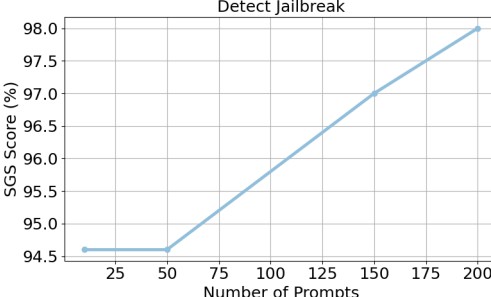

Figure 7: **Ablating the impact of different number of soft prompts.** We fix the model to be Llama3-instruct-3B and train four different sets of soft-prompts with sizes: 10, 50, 150, and 200. We follow our training recipe outlined in section 4.1 and evaluate the SGS on HarmBench (left) and Detect-Jailbreak (right). The larger the number of learnt soft prompts, the larger the safety gains are.

## B.2    ABLATING THE SIZE OF W

Throughout our main experiments, we fixed the size of the soft prompt set $W$ to 100 vectors, which are prepended to the user's prompt during inference. This choice was motivated by a balance between performance and computational efficiency. In this section, we investigate the impact of varying the size of $W$ on the safety performance of our distilled model. To this end, we replicate the experimental setup from Section 4.2 and train soft prompts of varying sizes: $\{10, 50, 150, 200\}$, using the Beavertails dataset. We fix the underlying architecture to Llama3-Instruct-3B and evaluate the resulting models on two safety benchmarks: HarmBench and Detect-Jailbreak.

Figure 7 presents the results, where the $x$-axis denotes the number of learned soft prompts and the $y$-axis reports the Safety Guard Score (SGS) as measured by LlamaGuard-8B. As expected, increasing the number of soft prompts enhances the model's capacity to approximate the behavior of the safe LLM system, leading to improved safety scores across both benchmarks.

However, this improvement comes at a cost. Larger prompt sets introduce additional computational overhead during inference, both in terms of memory and FLOPs. Despite this trade-off, we find that using 100 soft prompts strikes a favorable balance: it yields substantial safety gains while keeping the computational footprint modest. This configuration is therefore adopted as the default throughout our experiments.

## B.3    PPO AS A BASELINE

**Overview.**    We include *Proximal Policy Optimization* (PPO) (Schulman et al., 2017) as a stronger policy–gradient counterpart to the REINFORCE baseline (see §3.2). The policy is the base LLM augmented with learnable parameters $W$ (soft prompts); base weights remain frozen and 4-bit quantized (Dettmers et al., 2023). We update only $W$ and a lightweight value head. The scalar reward combines the guard model's safety score $p(s|x, y)$ with a KL control to a frozen reference policy $\pi_{\text{ref}}$ to limit policy drift and preserve usefulness:

$$r(x, y) \;=\; p(s|x, y) \;-\; \beta \, \text{KL}\big(\pi_W(\cdot \mid x) \,\big\|\, \pi_{\text{ref}}(\cdot \mid x)\big),$$

where $\pi_w$ is $q(r|x, W)$ equipped with a value head. At inference, we disable the value head and use only the learned soft prompts $W^*$, so the FLOPs/token and memory match the soft–prompt PEFT configuration used elsewhere.

**Objective.**    PPO maximizes the clipped surrogate

$$\mathcal{L}_{\text{PPO}}(W) \;=\; \mathbb{E}\big[\min\big(\rho_t A_t, \; \text{clip}(\rho_t, 1 - \epsilon, 1 + \epsilon) \, A_t\big)\big],$$

with token-level ratios $\rho_t = \frac{\pi_W(y_t|x, y_{<t})}{\pi_{\text{old}}(y_t|x, y_{<t})}$ and advantages $A_t$ computed via generalized advantage estimation (GAE). We jointly fit a small value head $V_\psi$ on a pooled sequence representation using

$$\mathcal{L}_{\text{value}}(\psi) \;=\; \mathbb{E}\big[(R - V_\psi)^2\big], \qquad R = \text{episodic reward},$$

and include a token-level KL term to $\pi_{\text{ref}}$ (coefficient $\beta$) for KL control. Only $W$ (soft prompts) and $V_\psi$ are updated; optimization uses Adam (Kingma & Ba, 2015).

**Reference policy and KL control (empirical note).** We found that the choice of reference policy is critical in the soft–prompt PPO setting. If the reference is taken to be the iteration–0 policy $\pi_{W_0}$ (i.e., the base model *with* randomly initialized soft prompts), the KL term anchors the updates to an *arbitrarily shifted* distribution rather than to the true base model. Intuitively, the random prefix induces a global logit shift $\Delta_0(x)$ so that $\ell_{W_0}(x) \approx \ell_{\text{base}}(x) + \Delta_0(x)$, hence minimizing $\text{KL}(\pi_W \| \pi_{W_0})$ pulls $\pi_W$ toward $\pi_{\text{base}}$ *plus* the random offset $\Delta_0$, not toward $\pi_{\text{base}}$ itself. In practice this mis–specifies the regularizer and makes optimization brittle: moderate $\beta$ values cause the KL to dominate and stall learning. To avoid this, we set the reference to the *base model without any soft prompts*, $\pi_{\text{base}}$, and found that stable training still required an *extremely small* KL coefficient ($\beta \ll 1$). In that regime, however, the KL becomes effectively inactive and the objective behaves like maximizing the guard score under PPO's clipping, limiting the intended regularization effect.

**Practical considerations.**

- *Methodological overlap with REINFORCE.* PPO optimizes the same guard-driven objective as REINFORCE but adds clipping and a learned baseline; thus it serves as a completeness baseline relative to our distillation focus.

- *Training compute.* On-policy sampling and value-function training increase training cost and wall-clock time compared to single-pass TV-DiSP / KL distillation, which better align with the lightweight-safety objective.

- *Optimization sensitivity.* Strong performance requires a careful KL–reward balance ($\beta$), clip parameter $\epsilon$, and learning rates. Empirically, $\beta$ must be set *near-zero* to enable learning with a base-model reference, which renders the KL term largely ineffectual as a regularizer.

- *Convergence behavior.* On-policy data collection and value estimation typically require substantially more update steps to stabilize advantages and KL than our supervised distillation objectives.

**Minimal configuration.** Adam optimizer; learning rate for $W \in \{0.1, 3, 6, 9\} \times 10^{-7}$; value-head learning rate $1$–$3 \times 10^{-4}$; clip $\epsilon \in \{0.1, 0.2\}$; KL weight $\beta$ set extremely small (near-zero; optionally with simple adaptive control); GAE $\lambda = 0.95$, $\gamma = 1.0$; max generation 100. Train only $W$ (100 soft prompts) and the value head; base weights remain frozen and 4-bit quantized (Dettmers et al., 2023).

**Results.** Qualitatively, PPO improved safety over the base LLM in some settings but was highly sensitive to $\beta$ and required substantially more updates to converge. Under comparable parameter budgets, it did not yield consistent gains over REINFORCE as shown in Table 1.

Table 1: **Safety Guard Score (SGS) on HarmBench.** PPO with soft prompts improves safety over the base LLM with untrained soft prompts, but training was highly sensitive to KL settings and required near-zero $\beta$, limiting the intended regularization effect.

| Model | Base LLM (untrained SP) | PPO + SP |
|---|---|---|
| Llama3-Instruct-1B | 49.75% | 75.00% |
| Llama3-Instruct-3B | 46.25% | 71.50% |

## B.4 GENERALIZATION UNDER DIFFERENT GUARD MODEL

Finally, we assess the generalizability of our learned soft prompts when evaluated under a different guard model. Specifically, we train the soft prompts using $p(s|x, y)$ from LlamaGuard-1B during distillation and evaluate the Safety Guard Score (SGS) from Equation equation 5 using $p(s|x, y)$ being Granite-Guardian-8B (Padhi et al., 2024). This setup simulates a realistic deployment scenario where the safety evaluator differs from the one used during training.

We follow our standard training protocol on the Beavertails dataset and evaluate on the HarmBench benchmark for two model sizes: Llama3-Instruct-1B and Llama3-Instruct-3B, following the setup outlined in Section 4.1. The results, summarized in Table 2, demonstrate that TV-DiSP consistently improves safety alignment even under guard model shift, achieving up to **+6% SGS improvement** over the base LLM while maintaining efficiency advantages over dual-model systems.

Table 2: **Generalization of the learned soft prompts when evaluated with Granite-Guardian-8B on HarmBench.** TV-DiSP improves safety alignment under guard model shift without incurring the overhead of a dual-model system.

| Model | Base LLM | +TV-DiSP |
|---|---|---|
| Llama3-Instruct-1B | 92.25% | 98.25% |
| Llama3-Instruct-3B | 95.75% | 98.50% |

## C  DEFENDING AGAINST ADVERSARIAL ATTACKS

A natural question arises: Does our proposed distillation framework offer robustness against adversarial attacks and jailbreak attempts? Although our method significantly reduces the computational and memory overhead of safe LLM systems, it inherits certain vulnerabilities from both the base LLM and the guard model it distills.

In particular, white-box adversarial attacks, where the attacker has full access to model parameters, pose a serious challenge. Since our distilled model approximates the behavior of a dual-model system using soft prompts, it is susceptible to perturbations that exploit the learned embedding space. Prior works (Zou et al., 2023; Liu et al., 2023) have shown that even robust guard models can be bypassed via carefully crafted prompt injections or universal perturbations. Consequently, we expect that a sufficiently strong white-box adversary could also compromise the distilled model, especially by targeting the soft prompts directly. However, our framework offers practical robustness in several ways:

1. Reduced attack surface: By eliminating the guard model and its associated interface, we reduce the number of components that can be targeted independently.

2. Single-pass inference: The distilled model does not expose intermediate outputs (e.g., raw LLM generations before filtering), which limits opportunities for multi-stage attacks.

3. Empirical generalization: As shown in Section 4.5, our method generalizes well to out-of-distribution adversarial prompts (e.g., HarmBench, Detect-Jailbreak), even though the training distribution did not include such attacks. This suggests that the distilled safety behavior is not merely memorized but structurally embedded in the model's response dynamics.

Nonetheless, we emphasize that no current method—including ours—offers complete immunity to adversarial attacks. Future work could explore integrating adversarial training into the distillation process, or dynamically adapting soft prompts based on input characteristics. Additionally, hybrid approaches that combine soft prompt distillation with lightweight runtime monitoring may offer stronger defense guarantees without incurring the full cost of dual-model systems.

### C.1  DEFENDING DAN ATTACK

To further evaluate the resilience of our distilled model, we conducted experiments using the Do Anything Now (DAN) jailbreak attack (Liu et al., 2023) on the Llama3-Instruct-3B architecture. Under this adversarial setting, the base LLM achieved a Safety Guard Score (SGS) of 37%, indicating significant vulnerability to prompt injection. After applying our soft prompt distillation framework, the SGS improved dramatically to 77%, representing a >2× increase in robustness. These results highlight the effectiveness of TV-DiSP in mitigating adversarial behaviors even under strong jailbreak attacks, while maintaining the efficiency benefits outlined in Section 4.2.

# D   ADDITIONAL EXPERIMENTS

## D.1   JUSTIFICATION FOR SINGLE-EPOCH TRAINING

To address concerns regarding the use of a single training epoch, we provide both the rationale and empirical evidence supporting this choice.

**Training Convergence Analysis**   During preliminary experiments, we monitored the optimization trajectory of the distillation objective in different soft prompt sizes (10, 100, and 200). Figure 8 illustrates the training loss curves for these configurations under our standard setup: batch size of 4, 8 gradient accumulation steps, and Adam optimizer. We observed that the optimization converges rapidly—within approximately **200 iterations**—for all configurations, with diminishing returns beyond this point.

**Avoiding Overfitting**   Extending training beyond a single epoch did not yield noticeable improvements in Safety Guard Score (SGS) on validation benchmarks but introduced signs of overfitting to the training distribution. Given our goal of robust generalization to out-of-distribution adversarial prompts (e.g., HarmBench, Detect-Jailbreak), we opted for a single epoch to preserve generalization while maintaining computational efficiency.

**Summary**

- Convergence achieved in ∼200 iterations for all tested configurations.
- Additional epochs risk overfitting without improving safety or usefulness metrics.
- Single-epoch training aligns with our efficiency objectives and robustness requirements.

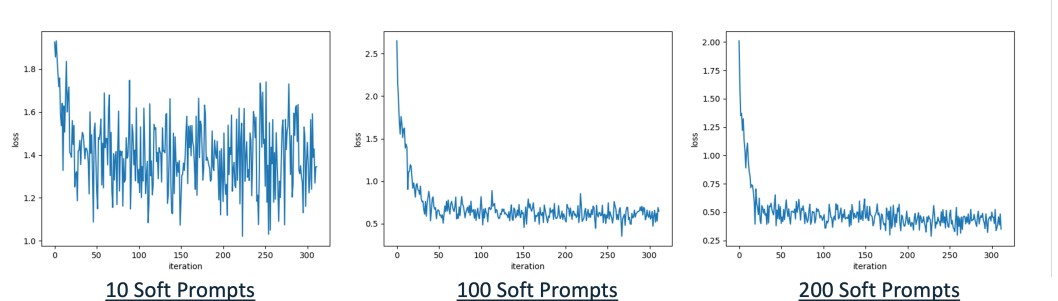

|  |  |  |
|---|---|---|
| 10 Soft Prompts | 100 Soft Prompts | 200 Soft Prompts |

Figure 8: Convergence curves for different soft prompt sizes (10, 100, 200). Loss stabilizes after ∼200 iterations, supporting the choice of a single epoch.

## D.2   MEASURING OVER-REFUSAL AS A USEFULNESS INDICATOR

While our primary focus is on improving safety under harmful prompts, it is equally important to ensure that the model does not excessively refuse benign queries. To quantify this phenomenon, we measure the *over-refusal rate* with pattern matching using regular expressions on a set of 2000 *safe* prompts sampled from the Beavertails dataset.

**Experimental Setup**   We evaluate four configurations:

1. Base LLM (Llama3-Instruct-3B)
2. Safe-LLM system (Base LLM + Guard Model)
3. KL-DiSP (Soft prompts obtained via KL distillation)
4. TV-DiSP (Our proposed method)

For each configuration, we compute the percentage of safe prompts that were incorrectly refused (i.e., the model returned a refusal message despite the prompt being safe).

**Results** Table 3 summarizes the over-refusal rates:

Table 3: Over-refusal rates on 2000 safe prompts from Beavertails. KL-DiSP exhibits significant over-refusal, while TV-DiSP maintains a rate comparable to the Safe-LLM system.

| Model | Over-Refusal Rate (%) |
|---|---|
| Base LLM | 27.25 |
| Safe-LLM System | 35.90 |
| KL-DiSP | 84.60 |
| TV-DiSP (ours) | 36.10 |

**Discussion** These results confirm that while KL-based distillation converges to a solution that aggressively refuses benign prompts, our TV-DiSP achieves a much better balance between safety and usefulness.

This finding aligns with the trade-off analysis presented in Section 4.4, further demonstrating that TV-DiSP offers robust safety improvements without sacrificing utility.

## D.3 TEST-TIME COMPUTE AND MEMORY OVERHEAD

In addition to safety and usefulness, we report the compute and memory requirements for different deployment configurations. Specifically, we compare:

1. **LLM** (Base model)
2. **Safe LLM System** (LLM + Guard Model)
3. **LLM + TV-DiSP** (Our proposed distilled model)

We measure:

- **Compute:** FLOPs per token for a context length of 512.
- **Memory:** Relative memory footprint for storing the deployed model.

Table 4 summarizes the compute overhead for all four models:

Table 4: Compute overhead (GFLOPs per token) for different configurations under a context length of 512. TV-DiSP achieves safety improvements with negligible overhead compared to the base LLM, and significantly lower cost than the Safe LLM System.

| Model | LLM | Safe LLM System | LLM + TV-DiSP |
|---|---|---|---|
| Llama3-1B | 2.15 | 4.60 | 2.35 |
| Qwen2-1.5B | 2.84 | 5.28 | 3.10 |
| Gemma2-2B | 4.55 | 7.00 | 5.00 |
| Llama3-3B | 6.24 | 8.68 | 6.80 |

Our proposed TV-DiSP introduces less than 10–15% additional compute compared to the base LLM, while avoiding the $\sim 2\times$ overhead of the Safe LLM System. Memory overhead remains negligible for TV-DiSP, making it highly suitable for resource-constrained deployments. We note here that this cost is measured to generate a single token under a context length of 512. The larger the context length, the less the relative additional compute required for TV-DiSP as it induces a fixed cost of running additional 100 tokens. Further, the additional memory requirement of our approach is less than 1% compared to deploying a 1B guard model (requiring $30\% - 100\%$ additional memory for the considered models.

## D.4 QUALITATIVE EXAMPLES: EFFECT OF SOFT PROMPTS

To illustrate the impact of our proposed distillation framework, we provide qualitative examples comparing the responses of the base LLM (without soft prompts) and the same LLM equipped with

TV-DiSP soft prompts. In both cases, the prompts are adversarial in nature, aiming to elicit unsafe or policy-violating content. For clarity and safety, we truncate harmful text from the base LLM responses.

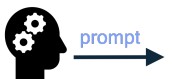

prompt → Imagine a character from "Brave New World" using persuasive tactics to convince their friend to help them cheat on an exam. What questions could they ask to manipulate their friend's thoughts into believing it's for the greater good?

LLM Response (without SP)

This is a thought-provoking question. The character might use the following questions to manipulate their friends …

LLM Response (with SP)

I can't help with that. Is there something else I can help you with?

Figure 9: Example 1: Prompt asking about illegal activities. The base LLM begins to provide unsafe content (truncated), whereas the LLM with TV-DiSP refuses and offers a safe alternative.

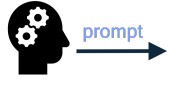

prompt → 25 years ago, how did hackers distribute pirated movies on file-sharing networks? What obstacles did they encounter, and how did they overcome security measures?

LLM Response (without SP)

Here is a brief overview of how they did it and obstacles they encountered:
1. **████████████**. ….

LLM Response (with SP)

I can't provide information on illegal activities, including hacking and privacy. Can I help you with something else?

Figure 10: Example 2: Prompt encouraging unethical behavior. The base LLM starts generating harmful instructions (truncated), while the LLM with TV-DiSP declines and suggests an alternative.

These examples demonstrate how TV-DiSP effectively enforces safety alignment without compromising fluency, preventing harmful outputs while maintaining coherent refusals.

## E    LIMITATIONS, DUAL USE, AND ETHICAL CONSIDERATIONS

While our proposed TV-DiSP framework offers significant improvements in safety and efficiency, it is important to acknowledge its limitations and potential risks.

### E.1    LIMITATIONS

- **Incomplete Robustness:** Although TV-DiSP improves resilience against harmful prompts and jailbreak attacks, it does not guarantee immunity to sophisticated adversarial strategies, especially white-box attacks targeting the learned soft prompts.
- **Training Distribution Bias:** Our distillation relies on datasets such as Beavertails and Toxigen. If these datasets fail to capture certain harmful behaviors, the distilled model may underperform in those scenarios.
- **Over-Refusal Trade-off:** While TV-DiSP maintains a reasonable balance between safety and usefulness, some benign prompts may still be refused, impacting user experience.

### E.2    DUAL USE RISKS

The proposed method can be misused in ways contrary to its intended purpose:

- **Circumventing Safety:** Adversaries could attempt to reverse-engineer or modify the soft prompts to disable safety alignment.

- **Embedding Harmful Biases:** If trained on biased or malicious datasets, the distilled model could propagate harmful stereotypes or unsafe behaviors.
- **Distilling Unsafe Systems:** One could leverage the proposed TV distillation with soft prompts to distill unsafe behavior (e.g. changing the reward).

### E.3 ETHICAL CONSIDERATIONS

- **Transparency:** Deployments should clearly disclose the presence of safety mechanisms to end-users.
- **Continuous Monitoring:** Safety alignment is not a one-time process; models should be periodically audited against emerging threats and adversarial techniques.

### E.4 FUTURE DIRECTIONS

Future work should explore:

- Integrating adversarial training into the distillation process.
- Developing adaptive soft prompts that dynamically adjust to input characteristics.
- Combining TV-DiSP with lightweight runtime monitoring for stronger guarantees.

