# OpenReview forum: "Distilling Safe LLM Systems via Soft Prompts"
_ICLR.cc/2026/Conference — Submitted to ICLR 2026_

### Official Review · Reviewer_F8Yb · 2025-10-24

**Soundness:** 3
**Presentation:** 3
**Contribution:** 2
**Rating:** 2
**Confidence:** 5

**Summary:**

This paper proposes TV-DiSP, a framework for distilling dual-model safety system, composed of a base LLM and a guard model, into a single model augmented with soft prompts. Instead of running a full guard model at inference time, the authors train soft prompt embeddings to minimize the total variation distance between the response distribution of the two-model system and that of the distilled single model. At test time, the learned soft prompts are prepended to user prompts, enabling the LLM to emulate the “safe” behavior of the guarded system with dramatically lower memory and compute overhead.

**Strengths:**

* Timely motivation: tackles a real deployment bottleneck—guard models are expensive for on-device inference.
* Simplicity: soft-prompt distillation is lightweight, easy to reproduce, and compatible with quantized models.
* Comprehensive experiments: includes multiple datasets, model sizes, and PEFT baselines.
* Balanced evaluation: assesses both safety (SGS) and usefulness (MMLU).
* Generalization results: shows modest gains even on out-of-distribution datasets like Detect-Jailbreak.

**Weaknesses:**

* Limited novelty. TV-based loss and prompt distillation are modest extensions over prior PEFT safety methods (GuardFormer, SafeLoRA, Safety-Prefix, etc.).

* Guard dependency. The method is entirely dependent on the guard model’s decisions, so any bias or blind spot in the guard is replicated in the distilled system.

* No ablations on number or location of soft prompts. Appendix B.2 provides limited analysis (10–200 prompts) but no clear convergence curves.

* Questionable evaluation metric. The “Safety Guard Score (SGS)” is derived from a different guard model (LlamaGuard 8B) and may not reflect true human-judged safety i.e llm as a safety judge might be incorrect/biased etc.

* No latency measurements (token/sec) or deployment case studies despite mention of edge-device motivation.

**Questions:**

1. How does TV-DiSP compare against supervised fine-tuning on guard-labeled data, without distillation?
2. Did you evaluate safety degradation over time (catastrophic forgetting) when mixing safety and general downstream data?
3. How sensitive is the method to the choice of guard model (e.g., LlamaGuard, Nvidia-Nemo Guard vs Granite Guardian)?
4. Can this approach compound safety errors if the guard model mislabels benign responses?
5. Would adding adversarial training or data augmentation improve robustness beyond imitation of the guard?

---

> ### Author Response · Authors · 2025-11-24
> **Rebuttal**
>
> We thank the reviewer for their insightful feedback. We are glad that the reviewer recognizes the importance of our study, the simplicity of our approach and the comprehensiveness of our experiments. Below, we provide a detailed response (along with additional experiments also in App. C-D) for each point in the weaknesses.
>
> (1) Regarding the limited novelty:
> We appreciate the reviewer's observation and would like to clarify the unique aspects of our work. While prior PEFT safety methods such as GuardFormer, SafeLoRA, and Safety-Prefix introduce parameter-efficient strategies for safety alignment, our approach differs in two key ways:
> (a) Unlike GuardFormer, SafeLoRA, and Safety-Prefix, which mainly steer toward refusal behavior, our approach distills the entire safe LLM system into the base LLM, preserving both safe-answer and refusal cases.
> (b) Probabilistic guarantees via TV distance: We introduce a total variation-based objective that provides formal bounds on downstream performance differences between the distilled model and the original safe system. To our knowledge, such guarantees have not been explored in prior PEFT safety work
>
> (2) Regarding the guard dependency:
> We agree that our method inherits the safety policy of the guard model, as it is designed to distill the behavior of a dual-model system. This is intentional: the goal is to replicate the safety guarantees of the deployed system in a lightweight form. Importantly, this does not introduce new bias as it reflects the same operational safety criterion already used in practice. To mitigate overfitting to a single guard, we (i) evaluate with a different guard model than the one used for distillation, and (ii) show generalization to out-of-distribution prompts (HarmBench, Detect-Jailbreak) and even under guard model shift (Granite Guardian), demonstrating robustness beyond the training guard. Future work could incorporate ensemble guards or human feedback to further reduce dependency.
>
> (3) Regarding the convergence curves: We thank the reviewer for this comment. We updated our appendix (refer to Appendix D.1) to include both discussions and convergence curves to address this point. This complements our analysis and ablations on the impact of different number of soft prompts presented in Appendix B4.
>
> (4) Regarding the use of guard models in evaluation:
> We acknowledge that human-judged safety is ideal, but large-scale human evaluation is infeasible for our setting. Following established practice in recent safety literature (such as HarmBench (Mazeika et al., 2024), JailbreakBench (Chao et al., 2024), and LlamaGuard (Inan et al., 2023)) we use a strong LLM-based guard model (LlamaGuard-8B) as an evaluator because it provides a reproducible and scalable proxy for safety. Importantly, we evaluate with a different guard model than the one used for distillation to reduce bias. Further, we also tested with different family of guard models (Table 2 reports results under Granite Guardian guard model). We also complement SGS with refusal rates and MMLU accuracy to capture both safety and usefulness.
>
> (5) Regarding the latency measurements: In Figures 2 and 6, the x-axis represents test-time compute (in Flops) to compare the latency of different approaches. Note that under different devices, the same number of flops can result in different runtime (in seconds). Nonetheless, we included in Appendix D.3 the detailed comparison in terms of computational requirements between different methods highlighting the efficacy of our TV-DiSP.
>
> (6) Regarding the comparison with supervised fine-tuning: Our "Perplexity" baseline is indeed the simple safety finetuning approach where conduct SFT on harmless data. We observe that while this approach can results in in-domain robustness, it can significantly overfit to the training distribution failing at generalizing to unseen domain (as shown in our experiments).
>
> (7) Regarding safety degradation: we thank the reviewer for raising this interesting point. In all our experiments, we focused solely on the safety aspect, thus our training only included safety data. We leave the exploration of including multi-objective distillation (safety + general downstream tasks) to future work is it falls outside the scope of our submission.
>
> (8) Regarding the sensitivity to different guard models: In Appendix B4, we analyzed the impact of distilling our soft prompts with one family of guard models (llama guard), and evaluate with another family of guard models (granite guardian). We observed that our soft prompts are not sensitive to the choice of the guard model providing consistent safety gains under different choices of guard models.
>
> (9) Regarding inheriting errors of distilled guard model: Indeed! as discussed in Appendix C, our approach distills the safe LLM system, thus inheriting the vulnerabilities from both the base LLM and the guard model, as explained in answer (2) as well.
>
> We are happy to engage in further discussions.

---

### Official Review · Reviewer_2MZs · 2025-10-26

**Soundness:** 2
**Presentation:** 3
**Contribution:** 3
**Rating:** 4
**Confidence:** 4

**Summary:**

This paper proposes TV-DiSP, a method that distills a dual-model safe LLM system (base model + guard) into a single model using soft prompts. The approach trains soft prompts by minimizing the total variation distance between the outputs of the safe system and the base model with soft prompts. This aims to retain safety while avoiding the compute and memory cost of a separate guard model. Experiments on several small instruction-tuned models show improved safety scores with limited overhead. The paper compares against alternative parameter-efficient tuning methods and loss objectives, and includes ablations on soft prompt size and a brief discussion of adversarial robustness.

**Strengths:**

**Strength 1**: The paper proposes soft prompts to approximate the behavior of a dual-model safe LLM pipeline, improving safety while requiring only minimal additional compute and memory, making it suitable for edge deployment.

**Strength 2**: The use of total variation as the distillation objective provides a clear performance deviation bound, contributing a sound theoretical foundation.

**Strength 3**: Experiments cover multiple base models, different training distributions, in- and out-of-distribution safety benchmarks, and comparisons with several PEFT and loss-based baselines.

**Weaknesses:**

**Weakness 1**: Both training supervision and evaluation rely on LlamaGuard models. The method may overfit guard model preferences instead of reducing real harmful behavior.

**Weakness 2**: The only measure of usefulness reported is 5-shot MMLU accuracy, which is a weak proxy for real-world utility in safety-critical contexts. Safety mechanisms may increase refusal rates or interfere with instruction following, but the authors do not report metrics that quantify harmful refusal or degraded responsiveness. As a result, the safety–utility trade-off remains unclear.

**Weakness 3**: Safety alignment requires high stability and low risk. Training only for a single epoch on a relatively small dataset does not provide evidence that the learned safety behavior is converged or robust. The reported improvements may not reflect a stable solution but rather transient behavior due to insufficient training.

**Questions:**

Q1: Could the authors provide justification for limiting training to a single epoch? Since safety alignment requires reliable and stable behavior, additional evidence such as performance curves across training steps would help verify that the reported safety gains reflect a converged solution rather than transient behavior.

Q2: Given the stated motivation of supporting efficient deployment, could the authors clarify why no Qwen3-series models were included?

Q3: Could the authors provide explicit numerical results for compute and memory efficiency in tabular format? The current figures give helpful intuition, but concrete numbers would allow clearer comparisons across models and methods, especially for deployment-oriented readers.

Q4: The discussion in Appendix Section C acknowledges the vulnerability of TV-DiSP to adversarial/jailbreak attacks but remains largely qualitative. Could the authors provide empirical results against recent prompt-based or embedding-level attack methods specifically targeting soft prompts? Even a small-scale evaluation would help quantify how much robustness is preserved compared to the dual-model system.

**Details Of Ethics Concerns:**

The method aims to replace a dual-model safety pipeline with a lightweight soft-prompt mechanism. While this improves efficiency, it also shifts the safety boundary into a small learned embedding space. If an attacker manages to bypass or suppress the influence of these soft prompts, the system may revert to unsafe behavior without the protection of a guard model. The paper currently provides limited empirical evidence on adversarial robustness, so the real-world deployment risk is not fully characterized. Additional validation would be needed before deployment in safety-sensitive scenarios.

---

> ### Author Response · Authors · 2025-11-24
> **Rebuttal**
>
> We thank the reviewer for their insightful comments. We are glad that the reviewer recognized the the efficacy of our proposed method, and soundness of our approach, and the comprehensiveness of our experiments.
> Below, we provide a detailed response (along with additional experiments) for each point in the weaknesses. Please, refer to Appendix C.1, and Appendix D in the updated paper for additional experiments and discussions.
>
> (1) Regarding overfitting to the guard model: We thank the reviewer for the comment. Our experiments in Appendix B.4 - Table 2) are precisely adressing this concern. In this experiment, we learnt our soft prompts with Llam3-Guard 1B and tested it with a different guard model family (Granite Guardian 8b). We observe that our approach is indeed generalizable under different guard models alleviating this weakness.
>
> (2) Regarding testing the over-refusing phenomena: We thank the reviewer for this important take. All our experiments (HarmBench, Detect-Jailbreak) are conducted on harmful prompts. This is since our work focuses on the safety concern of deployed LLMs. Nonetheless, we conducted experiments to measure the over-refusal rate on safe prompts.
> For this experiment, we test Llama3-instruct-3B on a set of 2000 *safe* prompts from Beavertails dataset. We measure the over-refusal rate of (i) Base LLM, (ii) Safe-LLM system, (iii) KL-DiSP (Soft prompts obtained via KL distillation), and our proposed TV-DiSP. The over-refusal rates are: 27.25\%, 35.9\%, 84.6\%, and 36.1\%, for the aforementioned models, respectively.
> This results shows that while KL-distillation converges to a solution that significantly over-refuses, our TV-DiSP results in a very comparable over-refusal rate to the safe LLM system. This confirms our results in Section 4.4 that our TV-DiSP offers a better tradeoff between robustness and accuracy efficiently.
>
> (3) Regarding the convergence under 1-epoch training: We also thank the reviewer for mentioning this point. The decision of conducting a single epoch is based on monitoring the training loss. We found that the optimization converges in 200 training iterations with batch size of 4 and 8 gradient accumulation steps. We refrained from multiple epochs of training to avoid overfitting the training data. We updated our appendix D.1 to include both discussions and convergence curves to address this point. This shows that our approach is both stable and low risk. Further, our experiments in Figures 3 and 4 shows the stability of our approach under different training hyperparameters strengthening the conclusion.
>
> (4) Regarding including results on Qwen3: We appreciate the reviewer's comment. In our paper, we experimented with 4 different LLMs including Qwen 2.5. We believe that this provides a comprehensive experimental validation for the proposed method.
>
> (5) Regarding reporting the compute and memory efficiency in tabular format: As per the reviewer's request, we dedicated a section in our appendix (Appendix D.3) with a corresponding table to explicitly mention the computation and memory requirements of different approaches.
>
> (6) Regarding experimentation with adversarial attacks: We thank the reveiwer for this suggestion. As per the reviewer's suggest, we conducted additional experiments to assess the generalizability of our learnt soft prompts via TV-DiSP to defend against Jailbreaks. In particular, we experimented with the standard DAN (Do Anything Now!) attack on Llama3-instruct-3B model on HarmBench dataset.
> We observe that, despite our soft prompts were not trained against DAN attacks, it is very effective in defending it.
> For example, our TV-DiSP enhances the SGS from 37\% for the base LLM to 77\% for the equipped version with our SP.
> This result strengthens the efficacy of our approach even under novel scenarios.
>
> We hope that our answers with the additional experiments provided addresses the reviewer's concerns. We are also happy to further engage in the discussion for any additional clarifications.

---

> > ### Comment · Reviewer_2MZs · 2025-11-25
> >
> > Thank you for the detailed response.
> >
> > However, my core concerns regarding utility degradation and adversarial robustness remain unresolved. I also note that similar concerns regarding the method's utility and soundness are shared by other reviewers.
> >
> > Therefore, I will maintain my score.

---

> > > ### Comment · Area_Chair_8njC · 2025-11-26
> > > **Ethics Assessment**
> > >
> > > Dear Reviewer, dear Authors,
> > >
> > > The Reviewer has raised an ethics concern here. After careful review of the paper, I think that this concern does not require a review by the Ethics Committee, as the concern is rather a "functional" one, asking for more evaluation, which has been provided by the Authors during the rebuttal. I would now kindly ask:
> > > 1. The authors to add a discussion section on limitations, dual use, ethical concerns of their method to the Appendix of the paper, discussing potential risks.
> > > 2. The reviewer to resolve their ethics flag.
> > >
> > > Thank you for your contributions to ICLR,
> > > The AC

---

> > > > ### Author Response · Authors · 2025-11-26
> > > > **Re Ethics Assessment**
> > > >
> > > > Dear AC,
> > > >
> > > > Thank you for your valuable feedback and for highlighting the importance of discussing limitations and ethical considerations. We have carefully addressed your request by adding a dedicated section titled “Limitations, Dual Use, and Ethical Considerations” in the appendix (Appendix E in the updated version of the paper of the revised manuscript). This section outlines:
> > > >
> > > > - Limitations of our approach, including incomplete robustness against sophisticated adversarial attacks and potential over-refusal trade-offs.
> > > >
> > > > - Dual-use risks, such as the possibility of misuse to circumvent safety mechanisms or embed harmful biases.
> > > >
> > > > - Ethical considerations, emphasizing transparency, continuous monitoring, and responsible release practices.
> > > >
> > > > - Future directions for mitigating these risks, including adversarial training and adaptive safety mechanisms.
> > > >
> > > > We believe this addition strengthens the paper by providing a balanced perspective on the societal and technical implications of our work. Please let us know if further elaboration should be included.

---

> > > ### Author Response · Authors · 2025-11-26
> > >
> > > Thank you for your follow-up. We appreciate your concerns and would like to clarify that we did address both points with additional experiments:
> > >
> > > - Utility degradation: We reported over-refusal rates on 2,000 safe prompts (Appendix B.4), showing that TV-DiSP maintains a refusal rate comparable to the safe LLM system (36.1% vs 35.9%), while KL-based distillation severely over-refuses (84.6%). This demonstrates that our method preserves utility significantly better than alternative safety tuning approaches.
> > >
> > > - Adversarial robustness: We included results on the DAN jailbreak attack (Appendix C.1), where TV-DiSP improves SGS from 37% (base LLM) to 77%, despite not being trained on adversarial prompts. This indicates strong generalization to unseen attacks.
> > >
> > > - Ethical considerations: As requested by the AC, we added a dedicated section in the appendix titled “Limitations, Dual Use, and Ethical Considerations” where we explicitly discuss potential risks, including dependency on the guard model, adversarial vulnerabilities, and mitigation strategies for deployment.
> > >
> > > We would be grateful for clarification on what additional evidence would address your concerns. We are happy to provide further experiments if specific directions are suggested.

---

> > > > ### Comment · Reviewer_2MZs · 2025-11-28
> > > >
> > > > Thank you for the detailed response and the additional experiments. They address many of my earlier technical questions, and the ethics concern is now fully resolved following the AC’s guidance and the updated appendix from the authors. However, based on the concerns raised by the other reviewers, I still see the conceptual novelty as incremental. The evaluation of usefulness and robustness lacks depth, especially considering the limited analysis of task-level utility, precision, recall, and broader adversarial coverage. Therefore, I keep my score. However, I wouldn’t strongly object if others rate the paper higher.

---

### Official Review · Reviewer_txFF · 2025-10-28

**Soundness:** 1
**Presentation:** 2
**Contribution:** 1
**Rating:** 2
**Confidence:** 4

**Summary:**

This paper proposes TV-DiSP, a safety distillation framework that replaces a dual-model safe LLM system (base model + guard model) with a single LLM augmented using learned soft prompts. The method aims to approximate the behavior of the safe system by minimizing a total variation distance (TVD) objective between the two output distributions. The distilled model produces refusal responses for unsafe prompts without requiring the guard model at inference, reducing both memory and compute costs. Experiments demonstrate improved Safety Guard Score (SGS) with significantly reduced inference overhead.

**Strengths:**

- The proposed framework is conceptually simple and offers improvements in computational efficiency compared to dual-model systems while maintaining competitive safety performance with regards to Safety Guard Score (SGS) only.
- Although TVD has been previously used in previous distillation works (e.g., [1]), this work presents empirical evidence that optimizing TV distance yields a better safety-utilization balance than KL divergence and REINFORCE on the benchmarks.

[1] Wen et al., f-Divergence Minimization for Sequence-Level Knowledge Distillation, ACL 2023

**Weaknesses:**

- The rationale for TVD is explained only briefly (lines 153-154) and remains general. Theorem 3.1 concerns generic distributional closeness guarantees, but the paper does not sufficiently demonstrate why TVD is inherently more appropriate for safety alignment than other divergences. If there is no clear reason, it seems just applying different distance metric in distillation (also covered by previous work [1]) instead of introducing the new method for safety tuning.
- Insufficient evaluation of model usefulness. Usefulness degradation is a critical part of safety alignment, yet the main safety-cost plots (Fig 2, 3, 4, 6) omit any utility measure. Only Fig 5 includes MMLU results, and it clearly reveals substantial performance drops compared to both the Base LLM and the Safe LLM system. There is also no deeper discussion of over-refusal and precision/recall or F1 over safety measures.
- Distillation versus supervised training is not justified. If safety labels are available (either human or synthetic), a supervised fine-tuning baseline would be natural comparison. It should be explained why distillation of the guard model’s imperfect predictions is preferable and what types of errors the proposed approach is expected to inherit.
- Comparison to alternative cost-reduction strategies is missing. If resource usage is the primary motivation, another reasonable baseline would be to distill large guard into a small guard and maintain a two-stage system [2]. This work does not discuss why shifting safety into soft prompts is preferable beyond efficiency, nor whether dual-model safety is fundamentally more robust.
- Notation clarity. Section 3 uses a single symbol $p$ for both LLM generation and guard scoring, without clarifying model parameters, which complicates understanding.

[1] Wen et al., f-Divergence Minimization for Sequence-Level Knowledge Distillation, ACL 2023

[2] Lee et al., HarmAug: Effective Data Augmentation for Knowledge Distillation of Safety Guard Models, ICLR 2025

**Questions:**

- In Sec. 3, do the base LLM and guard model share any parameters? If not, please distinguish them with separate parameter sets (e.g., $\theta$ for the LLM and $\phi$ for the guard).
- How is the distribution $p(r|x)$ represented in practice?
- Have you evaluated the model under metrics that separately measure precision vs recall of safe refusals?

---

> ### Author Response · Authors · 2025-11-24
> **Rebuttal**
>
> We thank the reviewer for their insightful comments. We are happy that the reviewer recognizes the computational efficiency of our proposed approach, and its efficacy in enhancing LLM's safety. Below, we provide a detailed response (along with additional experiments) for each point in the weaknesses. Please, refer to Appendix C.1, and Appendix D in the updated paper for additional experiments and discussions.
>
> (1)
> Regarding the rationale for TVD:
> We chose total variation distance because it provides symmetric, worst-case, guarantees on how closely the distilled model approximates the safe system. Theorem 3.1 shows that TVD directly bounds the worst-case difference in expected safety decisions, which is critical for safety-sensitive applications. In contrast, KL divergence is asymmetric and, depending on the direction, prioritizes high-probability regions of the reference distribution, which can bias optimization toward refusal and lead to over-refusal (as observed in Figure 5). TVD treats deviations in both directions equally, preserving both refusal and safe-answer behavior. Furthermore, as we discuss at lines 223-225 of the original draft, the KL-divergence acts as a looser upper bound (compared to TVD) for the worst-case gap that theorem 3.1 introduces. Finally, while prior work explored alternative divergences, none leverage TVD for safety alignment with formal guarantees. Our contribution thus lies in introducing this theoretically grounded objective and demonstrating its practical benefit in balancing safety and usefulness.
>
> (2) Regarding testing the over-refusing phenomena: We thank the reviewer for this important take. All our experiments (HarmBench, Detect-Jailbreak) are conducted on harmful prompts. This is since our work focuses on the safety concern of deployed LLMs. Nonetheless, we conducted experiments to measure the over-refusal rate on safe prompts.
> For this experiment, we test Llama3-instruct-3B on a set of 2000 *safe* prompts from Beavertails dataset. We measure the over-refusal rate of (i) Base LLM, (ii) Safe-LLM system, (iii) KL-DiSP (Soft prompts obtained via KL distillation), and our proposed TV-DiSP. The over-refusal rates are: 27.25\%, 35.9\%, 84.6\%, and 36.1\%, for the aforementioned models, respectively.
> This results shows that while KL-distillation converges to a solution that significantly over-refuses, our TV-DiSP results in a very comparable over-refusal rate to the safe LLM system. This confirms our results in Section 4.4 that our TV-DiSP offers a better tradeoff between robustness and accuracy efficiently. We included these results in Appendix D.2).
>
> (3) Regarding the supervised fine-tunning baseline:  Our "Perplexity" baseline is indeed the simple safety finetuning approach where conduct SFT on harmless data. We observe that while this approach can results in in-domain robustness, it can significantly overfiit to the training distribution failing at generalizing to unseen domain (as shown in our experiments).
>
> (4) Regarding alternative cost-reduction strategies: We completely agree with the reviewer. If the memory and computational requirments allow for a dual-model system, then leveraging guard models would be ideal. However, on many applications (e.g. on device setting) storing two models, and conducting two inferences for each received prompt might be infeasible. This is when approaches along the lines of the proposed one in this work shine.
>
> (5) Regarding the notation: In our notation, we used $p$ to present a generic probability distribution. For example, $p(y|x)$ is the probability distribution of outputs of an LLM, $p(r|x)$ is the probability distribution of the safe-LLM system's responses.
>
> (6) Regarding whether LLM and guard model share any parameters: No they do not.
>
> (7) Regarding the distribution of $p(r|x)$: Equation (2) in our submitted paper shows the mathematical formulation of $p(r|x)$. As discussed in lines 119-124, the output of the system will be the generation from the LLM if deemed safe by the guard model, otherwise will be a refusal message.
> For instance, if the received prompt is harmful, and the generation of the LLM contains harmful content, as judged by the guard model, a refusal message is returned.
>
> (8) Regarding evaluating safe refusals: All our experiments in the paper measured the SGS under harmful prompts. The additional results provided in the rebuttal regarding measuring over-refusal are conducted on harmless prompts.
>
> We hope that our answers with the additional experiments provided addresses the reviewer's concerns. We are also happy to further engage in the discussion for any additional clarifications.

---

### Official Review · Reviewer_dR65 · 2025-10-30

**Soundness:** 2
**Presentation:** 2
**Contribution:** 2
**Rating:** 2
**Confidence:** 4

**Summary:**

This paper presents an approach to distil an LLM-guard as additional prompts in the prompt embedding space to boost the LLM safety. Their approach works under the paradigm of a dual-model system, where after every generation, an LLM guard assesses the model's output to be harmful given the prompt and overrides the response with a fixed refusal response if it is harmful. Such systems can often be non-scalable due to prohibitive inference cost. Thus, they propose training additional input parameters in the prompt input embedding space to directly train the model to generate the refusal response based on the probability of the harm from the guard model. Experimental results show a tradeoff between improved safety of the generated responses and reduced compute, as assessed by the LLM-guard scores of the generations.

**Strengths:**

- The paper is mostly well-written with minimal typos and is easy to follow.
- It is well-motivated to address the inference cost of such classifier-based guardrails.
- Experimental results show that the inference computation remains similar to the original model while enhancing its safety.
- Both in and out-of distribution datasets are compared for comprehensiveness.
- Use of the guard model enables on-policy training and is shown by the use of PPO but is not fully highlighted.

**Weaknesses:**

- Performance gains are very sensitive to the guard model used, as Table 2 shows minimal gains as compared to Figure 1. Other metrics should be included that leverage the ground-truth labels of these.
- Performance should be split between harm-inducing and safe prompts separately to see if it is not overrefusing.
- Figure 5 shows little usefulness of the tuned model, which means the training is overrefusing and not able to clearly separate safe and harmful prompts.
- Since a simple safety finetuning can also have the same gains in reducing the inference cost and increasing the safety, the motivation behind the architectural design is not clear. Low performance of LoRA is not clear and should be further investigated in Figure 4.
- It is not clear why figures 4 and 5 show a trend with respect to the learning rate since only the best performance should be compared with the two trade-off metrics instead.
- Experimented models are limited in size (upto 3B models, quantized to 4 bit), leaving open questions about generalization to larger or unquantized models.
- Figure 6 should be extended to Toxigen and other LLMs.
- More discussion is needed regarding the use of many samples but only a single epoch for training.
- KL is shown to outperform the TV-DiSP method but it is not clear why the authors have still decided to go with the TV-DiSP loss.
- While it is highlighted that number of prompts can be tuned to hit a balance between the inference compute and safety, it should be clearly presented in the main figure by showing TV-DiSP with different number of prompts.
- In addition to the Llamaguard > 0.5 score, the exact average llamaguard scores and LLM judge based rejection evaluations of the generations should also be included, following existing works in the literature.
- Examples of generated responses are not included.
- It is not clear why existing benchmarks (JailbreakBench, StrongReject) are not used (with their jailbreaks). Furthermore, the details of the self-curated benchmark, DetectJailbreak, are not provided.
  - Chao, Patrick, et al. "Jailbreakbench: An open robustness benchmark for jailbreaking large language models." Advances in Neural Information Processing Systems 37 (2024): 55005-55029.
  - Souly, Alexandra, et al. "A strongreject for empty jailbreaks." Advances in Neural Information Processing Systems 37 (2024): 125416-125440.
- Missing related works: The architecture design is similar to Zheng et al., 2024 but is omitted from the discussion. Circuit breakers can also be seen as a fast fine-tuning strategy.
  - Zheng, Chujie, et al. "Prompt-driven llm safeguarding via directed representation optimization." ICML 2024.
  - Zou, Andy, et al. "Improving alignment and robustness with circuit breakers." Advances in Neural Information Processing Systems 37 (2024): 83345-83373.
- Minor:
  - Update the colors in Figure 1 right for better understanding.
  - Line 362: across all

**Questions:**

- What is the effect of changing the loss function to have binary labels of safe and unsafe as opposed to the probability? Safe can be detected as the guard model's probability or ground-truth in the dataset.
- see above weaknesses

---

> ### Author Response · Authors · 2025-11-24
> **Rebuttal (1/2)**
>
> We thank the reviewer for their insightful comments. We are glad that the reviewer found our work well motivated and well written, and recognized the comprehensiveness of our experiments. Below, we provide a detailed response (along with additional experiments) for each point in the weaknesses. Please, refer to Appendix C.1, and Appendix D in the updated paper for additional experiments and discussions.
>
> (1) Regarding the sensitivity to the guard model: Our learned soft-prompts are generalizing to different family of guard models. The results in Table 2 proves this point where the SGS, when measured with Granite Guardian is above 98\% under our learnt soft prompts. The main reason that the safety gain under the Guardian model is smaller is the fact that the guard model already finds the base LLM's generation quite safe (SGS > 90\%), unlike LlamaGuard 8B. This result, in fact, demonstrates that our approach is agnostic to the way it is tested.
>
> (2) Regarding testing the over-refusing phenomena: We thank the reviewer for this important take. All our experiments (HarmBench, Detect-Jailbreak) are conducted on harmful prompts since our work focuses on the safety of deployed LLMs. Nonetheless, we conducted experiments to measure the over-refusal rate on safe prompts.
> For this experiment, we test Llama3-instruct-3B on a set of 2000 *safe* prompts from Beavertails dataset. We measure the over-refusal rate of (i) Base LLM, (ii) Safe-LLM system, (iii) KL-DiSP (Soft prompts obtained via KL distillation), and our proposed TV-DiSP. The over-refusal rates are: 27.25\%, 35.9\%, 84.6\%, and 36.1\%, for the aforementioned models, respectively.
> This results shows that while KL-distillation converges to a solution that significantly over-refuses, our TV-DiSP results in a very comparable over-refusal rate to the safe LLM system. This confirms our results in Section 4.4 that our TV-DiSP offers a better tradeoff between robustness and accuracy. We included these results with extended discussion in Appendix D.2.
>
>
> (3) Regarding the usefulness in Figure 5: While indeed TV-DiSP reduces the MMLU accuracy, compared to the base-LLM, this reduction is the cost of the safety enhancement. We highlight here that TV-DiSP still remains the best performing method in terms of usefulness compared to other fine-tuning methods by a significant margin.
>
> (4) Regarding comparison with simple safety-finetuning: Our "Perplexity" baseline is indeed the simple safety finetuning approach doing SFT on harmless data. We observe that while this approach can result in in-domain robustness, it can significantly overfit to the training distribution failing to generalize to unseen domains (as shown in our experiments).
> Regarding why does the LoRA baseline exhibit low performance: We attribute this to two factors: (a) we set the rank for LoRA to match the number of learnable parameters of soft-prompts. This results in a very low rank (2-3) resulting in a limited capacity to learn the complex objective. (b) The weight quantization applied makes the optimization harder for the LoRA setting.
>
> (5) Regarding showing the trend of performance across different learning rates in Figures 4 and 5: The main purpose of this ablation is to show the consistency of the attained performance at several hyperparameters. This is an additional strength of our approach where it is robust against hyperparameters used during training.
>
> (6) Regarding the size of the tested models: In this work, we focused on on-device settings, with LLMs possibly deployed on the edge. In this setting, we are concerned with medium size models that are effective, but not memory and compute intensive ($<$8B parameters).
>
> (8) Regarding the discussion of the use of many samples and a single epoch: The decision of conducting a single epoch is based on monitoring the training loss. We found that the optimization converges in 200 training iterations with batch size of 4 and 8 gradient accumulation steps. We refrained from multiple epochs of training to avoid overfitting the training data. We updated our appendix to include both discussions and convergence curves to address this point (refer to Appendix D.1).
>
> (9) Regarding KL outperforming TV-DiSP: We respectfully disagree with the reviewer. While KL-based distillation can achieve Safety Guard Scores (SGS) comparable to TV-DiSP under its best hyperparameter setting, this comes at a significant cost to model usefulness. As shown in Figure 5, KL distillation causes severe over-refusal on harmless prompts, reducing MMLU accuracy by up to 20\% compared to TV-DiSP. In contrast, TV-DiSP achieves a better balance between safety and utility, preserving the base LLM’s capabilities while improving safety. This trade-off is critical for practical deployment, where excessive refusals degrade user experience.

---

> > ### Author Response · Authors · 2025-11-24
> > **Rebuttal (2/2)**
> >
> > (10) Regarding showing in the main figure TV-DiSP with different number of soft-prompts: We thank the reviewer for this suggestion. We intentionally made Figure 2 include results with a single choice of 100 prompts to better convey the message: our approach adds small computational overhead and enhances the safety significantly. We left the ablation over different number of soft prompts to the appendix (Figure 7).
> >
> > (11) Thank you for the suggestion. We did not report the exact LlamaGuard scores as our evaluation focuses on behavior with respect to the safety decision threshold (commonly 0.5), which is standard for downstream performance. As per the reviewer's request, we included in Appendix D experiments measuring over-refusal rates for different baselines.
> >
> > (12) Regarding including generated responses in our paper:
> > We updated our appendix to include examples of generated responses (please refer to Appendix D4).
> >
> > (13) Regarding experiments on JailBreakBench: We experimented with prompts from this benchmark and found that the base LLM, without any additional fine-tuning generates safe responses (SGS > 90\%). We focused on more challenging benchmarks where the base LLM is susceptible to generate harmful content when prompted from these datasets. Regarding Detect-Jailbreak: We note we did not curate this benchmark ourselves, but rather followed the standard practice (refer to the dataset's model card for more information https://huggingface.co/datasets/GuardrailsAI/detect-jailbreak?not-for-all-audiences=true). We highlight here that our experiments match, if not exceed, the complexity of experiments in published works in top-tier conferences such as the pointed out work of Zheng et.al [A]).
> >
> > (14-15) Regarding the missing related works and the typo: We updated our paper to correct the typo and include both works mentioned by the reviewer. We thank the reviewer for their thoroughness and pointing out these related references.
> > The related work investigated only refusal direction, while our work is meant to for distilling the safe LLM-system as a whole where the refusal is one case. Further, the mentioned work relies on labeled data through which a binary classifier is applied, unlike our work that operates in a self-supervised manner.
> >
> > (16) Regarding changing the loss function to have binary lables: We thank the reviewer for this great suggestion. Considering a binary label in the form (1/0) will simplify Equation (4) to optimize the soft prompts only on generations that are considered safe. This will not allow the soft-prompts to distill the behavior of the system under unsafe LLM generations.
> >
> > [A] Prompt-driven llm safeguarding via directed representation optimization, ICML 2024.
> >
> > We hope that our answers with the additional experiments provided addresses the reviewer's concerns. We are also happy to further engage in the discussion for any additional clarifications.

---

### Meta-Review · Area_Chair_8SYT · 2026-01-11

**Summary:**

This paper proposed a framework to use a single LLM to realize safety by distilling a guard model. The idea relies on that the distilled additional prompts in the prompt embedding space can enhance the LLM safety. Then, this method approximates the behavior of the safe LLM by minimizing a Total Variation Distance (TVD) between the two output distributions. Evaluation results show a tradeoff between outputs' safety and the decrease of the computation, as assessed by the LLM-guard scores of the generations. The paper received four negative reviews and there are many questions from the reviewers. There is only one reviewer engaged the rebuttal but mentioned about maintaining the score.

**Reviewer Concerns:**

- **Limited novelty.** TVD distillation + soft prompts; limited difference with PEFT/safety prompting approaches
- **Limited evaluation.** The results are largely proxied by MMLU and/or not shown in the main tradeoff figures, limited analysis of instruction-following quality and real “helpfulness” under safety constraints.
- **Over-refusal/safety-utility tradeoff not convinced.** Missing precision/recall-type safety metrics, limited safe-prompt evaluation, many claims pushed to appendices.
- **Robustness gaps.** limited coverage of jailbreak/adversarial methods specific to soft prompts, “guard removal/bypass” risk is acknowledged but not evaluated and convinced.
- **Dependence on guard models.** This issue exists for both supervision and evaluation raises concern about inheriting guard biases/errors and whether improvements reflect real safety.

**Reviewer Scores:**

There are four negative reviews for this paper. Only reviewer 2MZs engaged the discussion and mentioned about maintaining the score. There are no other reviewer's replies. After reading other reviewers' questions and rebuttals, my summary is as follows.
- For reviewer dR65's many weakness and questions, the authors partially addressed them but for the question about the "binary safe/unsafe labels vs guard probability—effect on loss/training?", the authors only gave some conceptual answer only without experiments to prove.
- For reviewer txFF's comments, the authors also did not give any further experiments especially considering the last two questions.
- For reviewer 2MZs's comments, the reviewer engaged in the discussion and claimed no change about the score since "this paper 's conceptual novelty as incremental, the evaluation of usefulness and robustness lacks depth, especially considering the limited analysis of task-level utility, precision, recall, and broader adversarial coverage."
- For reviewer F8Yb's comments, there are also several major questions remained unsolved. For instance, the question of "Sensitivity to guard model choice (LlamaGuard vs Granite Guardian etc.)" is not addressed with quantified results.

---

### Decision · Program_Chairs · 2026-01-26

Reject